# An Evaluation of Serum miRNA in Renal Cell Carcinoma: A Systematic Review

**DOI:** 10.3390/cancers17050816

**Published:** 2025-02-26

**Authors:** Giovanni Cochetti, Liliana Guadagni, Alessio Paladini, Miriam Russo, Raffaele La Mura, Andrea Vitale, Eleonora Saqer, Paolo Mangione, Riccardo Esposito, Manfredi Gioè, Francesca Pastore, Lorenzo De Angelis, Federico Ricci, Giacomo Vannuccini, Ettore Mearini

**Affiliations:** Urology Clinic, Department of Medicine and Surgery, Santa Maria della Misericordia Hospital, University of Perugia, 06129 Perugia, Italy; giovanni.cochetti@unipg.it (G.C.); liliana.guadagni@specializzandi.unipg.it (L.G.); miriam.russo@specializzandi.unipg.it (M.R.); raffaele.lamura@specializzandi.unipg.it (R.L.M.); andrea.vitale@specializzandi.unipg.it (A.V.); eleonora.saqer@specializzandi.unipg.it (E.S.); paolo.mangione@specializzandi.unipg.it (P.M.); riccardo.esposito@specializzandi.unipg.it (R.E.); manfredi.gioe@specializzandi.unipg.it (M.G.); francesca.pastore@specializzandi.unipg.it (F.P.); lorenzo.deangelis@specializzandi.unipg.it (L.D.A.); federico.ricci@specializzandi.unipg.it (F.R.); giacomo.vannuccini@unipg.it (G.V.); ettore.mearini@unipg.it (E.M.)

**Keywords:** microRNA, miRNA, renal cell carcinoma, biomarker, liquid biopsy

## Abstract

In the last decade, the role of microRNAs (miRNAs) as oncological biomarkers has been investigated, resulting in majorly heterogeneous outcomes. This systematic review aims to gather information about the diagnostic power of serum miRNAs in renal cell carcinoma (RCC) to help researchers navigate the vastness of the topic and to develop standardized protocols for future research to consolidate international findings.

## 1. Introduction

Renal cell carcinoma (RCC) represents around 3% of all cancers and is the most common solid lesion within the kidney. Due to their asymptomatic course, especially in the early stages, about 60% of RCCs are identified incidentally by abdominal US or CT. When symptoms begin to appear, the disease is often in advanced stage. About 20% of RCCs are metastatic disease at diagnosis. The prognosis of RCC largely depends on the histological type, clinical stage and grade [1].

In recent years, research has highlighted the importance of microRNAs (miRNAs) in the context of oncological diseases, including renal carcinoma. miRNAs are small non-coding RNA molecules, typically 18–25 nucleotides long, that regulate post-transcriptional gene expression by binding to complementary sequences on target mRNAs, leading to their degradation or translation inhibition. In renal carcinoma, miRNAs play a crucial role in regulating biological processes such as cell proliferation, apoptosis, angiogenesis and metastasis [2].

Some miRNAs are found to be deregulated in RCC, and their abnormal expression can contribute to carcinogenesis. Serum miRNAs, released into the bloodstream by tumor cells, offer unique opportunities for non-invasive diagnosis, monitoring disease progression and assessing treatment response. These miRNAs can be detected in various body fluids, including blood, urine and saliva, using advanced laboratory techniques such as quantitative PCR (qPCR), next-generation sequencing (NGS) and miRNA arrays. One of the main advantages of using serum miRNAs as biomarkers is their stability in the blood, making them ideal for repeated analyses over time [3].

Studies have shown that specific serum miRNAs can discriminate between healthy individuals and patients with renal carcinoma, as well as between different stages and grades of the disease. Despite its promising potential, the clinical use of serum miRNAs in renal carcinoma still faces several challenges. One of the main challenges is the standardization of methods for detecting and quantifying serum miRNAs. Differences in isolation protocols, sample storage and analytical techniques can lead to variable and difficult-to-reproduce results. Moreover, interpreting data on serum miRNAs requires a thorough understanding of their biological functions and interactions with other components of the tumor microenvironment.

The aim of this systematic review of the literature was to analyze the main serum miRNAs involved in RCC and their potential diagnostic power.

Our primary research question (RQ) is as follows: which serum miRNAs are differentially expressed in adult patients with RCC compared to healthy individuals?

Our secondary RC is the following: which serum miRNAs express different behavior before and after surgery in RCC patients?

## 2. Material and Methods

We performed the systematic review following the PRISMA (Preferred Reporting Items for Systematic reviews and Meta-Analyses) 2020 guidelines [4,5], integrated with the Synthesis Without Meta-analysis (SWiM) checklist [6] (Appendix A). The protocol was registered in PROSPERO (CRD42024550709) [7].

The inclusion criteria adhering to the PICO framework (Population, Intervention, Comparator, Outcome) were as follows:
Population: Adult (≥18 years old) patients with RCC.Intervention: Measurement of circulating or cell-free miRNA in blood samples of patients with RCC (exclusion: snRNA, ccRNA, exosomal RNA, lncRNA).Comparator/Control: Healthy subjects or patients with RCC after surgery.Outcome (main): Different expression of miRNA in blood samples between patients with RCC and healthy subjects through diagnostic accuracy measurements.

Outcome (additional): Different expression of miRNA in blood samples of patients with RCC before and after surgery through diagnostic accuracy measurements.

We included prospective cohort studies, randomized controlled trials, cross-sectional studies, case–control studies and case series.

The exclusion criteria were as follows:
Pediatric patients and adult patients with benign renal tumors.Measurement of RNA other than circulating or cell-free miRNA in blood samples of patients with RCC (snRNA, ccRNA, exosomal RNA, lncRNA…).Reviews and meta-analyses, abstracts, letters and meeting reports.

Literature search strategies were developed, composing strings with text words related to serum or blood miRNA in patients with RCC for PubMed, EMBASE and Clinicaltrial.gov.

Bibliographic citations were imported into the software Mendeley Desktop [8]. Then, the references were exported into a Microsoft Excel spreadsheet, which was used for study selection [9].

The study selection process was performed by three independent review authors (L.G., F.P., P.M.) and was conducted in two phases. In the first phase, the reviewers assessed the records through their titles and abstracts according to the inclusion criteria. Any disagreement was solved through discussion and, when necessary, a supervisor (G.C., A.P.) was involved. In the second phase, the other four review authors screened the full texts of the potentially eligible studies.

Data extraction was performed by eleven independent reviewers (L.G., F.P., M.R., E.S., A.V., R.L.M., F.R., P.M., M.G., L.D.A., R.E., G.V.) using a standardized table. To ensure consistency across the reviewers, they were thoroughly instructed on how to fill the table. Disagreements on extracted data were solved through discussion involving a supervisor (G.C., A.P.).

The following information was extracted from the included studies: bibliographic data (first author, publication year and citation), study characteristics (study design, country, number of centers, sample size), participant characteristics (disease, gender, age), intervention characteristics (type of miRNA, dosage method, normalizers and phase), comparator characteristics (healthy subjects or patients with RCC in other phases) and study outcomes.

We decided not to perform a meta-analysis because we expected excessive heterogeneity between studies in terms of type of miRNA, extraction method and normalizers used.

Mean difference (MD) was used to aggregate continuous data, while pooled risk ratio (RR) was used to aggregate dichotomous data. We used the inverse variance approach and the random effect model. We considered 95% confidence intervals.

Two independent reviewers (L.G., M.G.) conducted the risk of bias assessment after being given explicit instructions on how to use the Cochrane Risk of Bias Tool (RoB 2) for randomized controlled trials and the ROBINS-E (Risk Of Bias In Non-Randomized Studies—of Exposure) tool for non-randomized studies [10]. The records were categorized as low-risk, some concerns, high-risk or very-high-risk following a pilot phase to guarantee a uniform review.

One reviewer (L.G.) performed the certainty assessment using the GRADE (Grading of Recommendations Assessment, Development and Evaluation) approach [11]. During the inconsistency evaluation phase, we analyzed the publications separately rather than grouping them because there was a high discrepancy rate. The records were divided into four quality categories: high, moderate, low and very low.

## 3. Results

### 3.1. Study Selection

Among the three databases we used, 500 records were found. Following the elimination of duplicates, we reviewed 385 entries, examined 80 full-text documents, and ultimately included 26 studies [12,13,14,15,16,17,18,19,20,21,22,23,24,25,26,27,28,29,30,31,32,33,34,35,36,37] (Figure 1, PRISMA 2020 flow diagram). A list of excluded studies along with their reasons for exclusion is provided in Appendix A. The main reason for exclusion was a wrong outcome (35 studies). Other reasons for exclusion were a wrong study design, wrong comparator, no English version of the article existing and full-text documents not being available; one retired article was also excluded.

We did not carry out meta-analyses since the included studies were not homogeneous in terms of studied miRNA, the normalizer used during the stabilization phase or the type of RCC (clear cell RCC, papillary RCC, not-specified RCC). We described the main characteristics and findings of the included studies in Table 1. All the included records were diagnostic accuracy studies.

### 3.2. Key Findings

All researchers used synthetic or endogenous controls to stabilize the miRNAs after serum extraction. The most common normalizer used was cel-miR-39. A summary of all laboratory findings regarding extraction, normalization and quantitative analysis methods can be found in Appendix A.

As shown in Figure 2, miR-210 turned out to be one of the most studied circulating miRNAs in RCC. Four studies [14,19,24,37] found that miR-210 was overexpressed in RCC patients compared to controls, while Wulfken et al. [34] sustained that there was no difference in the expression of miR-210 between cases and controls. Iwamoto et al. [19] pointed out that miR-210 is also especially present in the early stages of the disease. Kalogirou et al. [20] studied a variant of the same family, miR-210-3p, and found no significant difference between RCC patients and healthy controls.

The miRNA that showed the greatest discrepancy among the studies was miR-378: Hauser et al. [15] showed a not-significant difference in miR-378 expression among RCC, benign tumors, and their control group, while others [14,27] showed an increase in miR-378 in patients with RCC compared to a healthy control group. Instead, Wang C. et al. [31] found it underexpressed in clear cell RCC (ccRCC) cases.

The expression of some miRNAs such as miR-1-3p and miR-129-5p was consistently lower in cases compared to healthy controls among the studies [23,33]. MiR-221, miR-222, miR-224-5p and miR-1233 were consistently overexpressed in RCC compared to controls, such as in [17,23,28,29,34,35].

In addition to their diagnostic role, some miRNAs showed associations with disease stage or grade. For example, miR-122-5p, miR-206 and miR-21-5p were linked to advanced-stage RCC [16,20], while Huang et al. found that higher miR-196a-5p levels correlated with lower Fuhrman grades [18].

Some authors such as Chen et al. (miR-21–5p, miR-150–5p, miR-145–5p and miR-146a-5p), Li R. et al. (miR-18a-5p, miR-181b 5p, miR-138-5p and miR-141-3p), Wen et al. (miR-1-3p, miR-129-5 and miR-146b-5p) and Wang C. et al. (miR-193a 3p, miR-362, miR-572, miR-28-5p and miR-378) evaluated panels from 3 to 5 miRNAs that may be useful in RCC screening [13,22,31,33].

Five papers reported different expressions in serum levels of miRNAs before and after surgery: miR-106a, miR-144-3p and miR-210 levels decreased after nephrectomy, while miR-22 levels increased [14,21,26,30,37].

### 3.3. Risk of Bias and Certainty Assessment for Included Studies

The methodological quality was assessed using the ROBINS-E tool [10] (Table 2) since all the included studies were non-randomized diagnostic accuracy studies.

During the critical appraisal phase, the overall risk of bias included 7 low-risk studies, 17 articles with some concerns of risk and 2 high-risk studies.

Most studies presented some concerns because no confounding factors were mentioned. Li M. et al. [21] had a high risk of bias due to the lack of methods reported on the extraction and treatment of serum miRNA. Lou et al. [26] had a high risk of bias due to the lack of explanation for miRNA selection during the validation phase.

The certainty assessment phase was performed using the GRADE method (Table 2) [11]. Since every study measured miRNA serum levels objectively through highly specialized laboratory equipment, non-randomized case–control studies were considered high-quality.

Due to their focus on potential RNA-based diagnosis, studies with a minimum of 200 participants between patients and controls in every study phase were deemed sufficient for numerosity.

The evaluation of imprecision in 11 studies was considered serious due to a small cohort of participants. Wang X. et al. [32] also lacked reports of diagnostic accuracy measures such as sensitivity and specificity and was deemed to be very serious for imprecision and very-low-quality.

Li M. et al. [21] work was considered serious during the evaluation of indirectness because it was mainly focused on miRNA’s expression in tissue samples instead of serum. The quality of this original article was also very low.

Out of 26 studies, only 3, Chen et al. [13] Fedorko et al. [14] and Teixeira et al. [29], were considered high-quality. We classified the quality of 13 articles as moderate, while 8 studies were considered low-quality.

## 4. Discussion

This systematic review, encompassing 26 studies, provides a detailed examination of circulating miRNAs as potential diagnostic biomarkers for RCC. The included studies exhibited marked heterogeneity in terms of their design, target miRNAs and RCC subtypes studied (ccRCC, pRCC or unspecified). Discrepancies can also be found in laboratory investigations, such as different methods of miRNA extractions, normalizers used and quantitative analysis.

This variability significantly limited the comparability of findings and the potential for quantitative synthesis. Among the investigated miRNAs, miR-210 emerged as a central focus, with most studies demonstrating its overexpression in RCC patients compared to controls.

Conversely, conflicting results for miRNAs such as miR-21-5p and miR-378, which showed limited or inconsistent differentiation between RCC and controls, highlighted the variability in miRNA behavior across different studies and settings. This inconsistency may reflect differences in study populations, disease stages or technical approaches, emphasizing the need for standardized protocols.

Many studies did not find any relevant expression of miRNAs in patients’ serum. This could be due to miRNA fluctuations or laboratory equipment not being able to detect lower blood concentrations of miRNAs.

Several studies proposed miRNA panels to enhance diagnostic accuracy [13,22,31,33]. These authors identified multi-miRNA combinations that could outperform individual biomarkers in sensitivity and specificity. Such panels may address the limitations of single-marker variability and improve robustness in diverse patient populations.

Some papers evaluated dynamic changes post-surgery, suggesting miRNAs’ potential utility as markers for both diagnosis and treatment monitoring. Our findings suggest that specific miRNAs could aid not only in diagnosis but also in assessing disease progression and guiding therapeutic decisions.

This systematic review represents the most comprehensive and up-to-date analysis of circulating miRNAs as diagnostic biomarkers for RCC to date. It builds upon and expands the scope of earlier reviews by incorporating a larger number of studies, reflecting the growing body of research in this field. In contrast to previous reviews, which often included a broader spectrum of RNA types or focused on tissue-derived miRNAs, our analysis exclusively examines serum-derived circulating miRNAs [38,39]. Similarly to miRNAs expressed in tissue, the dysregulated expression of serum miRNA might play a role in the generation or maintenance of solid tumors [40]. While investigating miRNAs expressed in tissue could help better understand their role in cancer generation, proliferation or metastatic processes, serum miRNA could offer a viable non-invasive diagnostic marker for earlier cancer detection.

A key strength of our review is the inclusion of a rigorous methodological tools, including a risk of bias assessment and certainty of evidence evaluation, approaches absent in prior reviews. By employing these frameworks, this review offers a more transparent and nuanced interpretation of the available data, identifying both promising candidates and critical gaps in the evidence base. Consistent with previous reviews, our analysis identifies miR-210 as a consistently overexpressed biomarker in RCC, reinforcing findings from earlier studies [38]. However, discrepancies among individual studies underscore the need for standardized protocols and larger validation cohorts to address variability in study designs and populations. Through a variety of processes, microRNAs control the expression levels of other genes, such as elevated levels of miR-210 occurring in response to hypoxia-inducible factors and their association with the hypoxia pathway. Additionally, miR-210 overexpression in tissues affected by heart illness and tumors (adrenocortical carcinoma, breast cancer and pancreatic, neck and head cancer) is known to be associated with a decreased chance of survival [41]. Moreover, miR-210 has been investigated for its potential to restore heart function following myocardial infarcts by inhibiting cardiomyocyte death and upregulating angiogenesis [42]; this could further suggest miR-210’s role in tumoral neoangiogenesis.

While previous studies primarily focused on individual miRNAs, our review also included studies evaluating the diagnostic role of miRNA panels, emphasizing the value of combining multiple markers to address their inherent variability and improve sensitivity and specificity.

However, our findings align with prior reviews in acknowledging that no circulating miRNA has yet achieved validation for routine clinical use. This reinforces the need for continued research, particularly in standardizing methodologies, addressing confounding factors and validating findings in larger, independent cohorts.

For this purpose, we identified the following critical gaps which should be addressed to improve future research on miRNAs’ diagnostic role:Validation in larger cohorts: The diagnostic utility of promising miRNAs and multi-miRNA panels must be validated in larger, independent cohorts to ensure their generalizability and clinical applicability.Standardization of methods: It is essential to develop standardized protocols for sample collection and data reporting to improve comparability between studies and increase the reliability of results.An international consensus on laboratory investigations for miRNA extraction, profiling, stabilization and quantitative analysis could provide a clearer interpretation of results.The institution of a global updated library of miRNA could help researchers explore not-yet-investigated miRNAs and consolidate international findings.Exploration of prognostic and therapeutic value: Further studies are needed to clarify the role of miRNAs in predicting disease progression, therapeutic response and patient outcomes.Integration of multi-miRNA panels: Combining multiple miRNAs into diagnostic panels may enhance accuracy and reliability, particularly for early detection and differentiation of RCC subtypes.Integration with other biomarkers: Integration of miRNAs with other biomarkers, such as protein or genetic biomarkers, could improve diagnostic precision.Assessment of confounding factors: Identifying and controlling for potential confounding factors is essential to minimize bias risk.Technological advances: Innovations in miRNA detection, including next-generation sequencing and machine learning-based analysis, could improve sensitivity and specificity, making miRNA-based diagnostics more viable in routine clinical settings.

The main limitations of this review included the limited number of databases considered for study selection and our decision to evaluate only the miRNAs involved in RCC studies, excluding potential associations with other histological types.

Although our review was in line with the current state of available evidence, considering the large number of existing miRNAs, a few papers on different miRNAs are not enough to determine strong biomarkers for RCC. We suspected that this number could be underestimated because of publication bias.

Moreover, a meta-analysis could provide a better understanding of potential biomarkers for RCC once a higher homogeneity among studies has been reached.

## 5. Conclusions

Circulating miRNAs represent a promising avenue for the non-invasive diagnosis and management of RCC. Our findings revealed substantial heterogeneity among the studies, both in the specific miRNAs investigated and the methodologies employed. Despite this variability, miR-210 emerged as differentially expressed in RCC patients compared to healthy controls. This review offers valuable insights into the current landscape of research while highlighting both the challenges and opportunities in utilizing miRNAs as biomarkers for RCC diagnosis. Future research should focus on standardization, validation in larger cohorts and the development of multi-marker diagnostic panels to address these current limitations and pave the way for miRNA-based diagnostics in RCC.

## Figures and Tables

**Figure 1 cancers-17-00816-f001:**
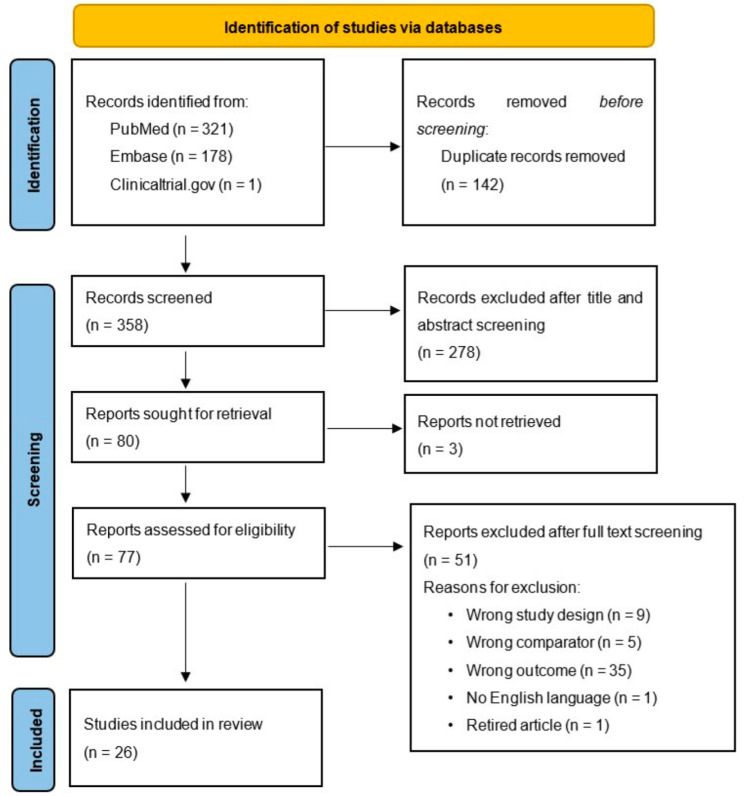
PRISMA 2020 flow diagram.

**Figure 2 cancers-17-00816-f002:**
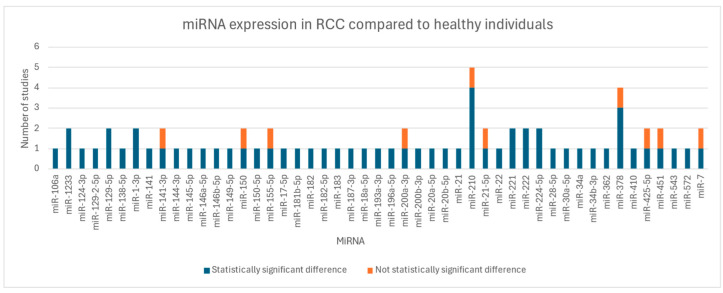
Summary of miRNAs expressed in our selection of studies.

**Table 1 cancers-17-00816-t001:** Included studies, main characteristics and summarized results. **RCC** (renal cell carcinoma); **ccRCC** (clear cell RCC); **pRCC** (papillary RCC); **chRCC** (chromophobe RCC); **nccRCC** (non-clear cell RCC); **sRCC** (sarcomatoid RCC); **↑** (overexpressed); **↓** (underexpressed).

Study (First Author, Publication Year, Country)	Number of Participants (Cases; Controls)	miRNA	Results
Chanudet E. (2017, France) [12]	194 (94 ccRCC; 100 controls)	288 miRNAs evaluated	miR451 + miR26b# discriminate cases from controls (AUC = 0.64); 60 miRNAs significantly differentially expressed in late-stage ccRCC compared with controls (miR-451 ↓ in late-stage ccRCC vs. controls); no significant differences in miRNA between early-stage ccRCC cases and controls. As prognostic factors, miR-150 and miR-587 ↑ in RCC survivors (q = 0.004 and q = 0.03, respectively)
Chen X. (2021, China) [13]	241 (123 RCC; 118 healthy)	30 miRNAs evaluated; 6 miRNAs (miR-145-5p, miR-146a-5p, miR-150-5p, miR-21-5p, miR-17-5p and miR-20a-5p) went through validation phase	miR-150-5p (*p* < 0.001) and miR-21-5p (*p* < 0.001) ↑ in RCC compared to controls; miR-145-5p (*p* < 0.001), miR-146a-5p (*p* < 0.001), miR-20a-5p (*p* < 0.001) and miR-17-5p (*p* = 0.004) ↓ in RCC compared to controls
Fedorko M. (2015, Czech Republic) [14]	295 (195 RCC; 100 healthy)	miR-378 and miR-210	miR-378 (*p* < 0.0001) and miR-210 (*p* < 0.0001) ↑ in RCC compared to controls; miR-378 (*p* < 0.0001) and miR-210 (*p* < 0.0001) ↓ in patients 3 months after nephrectomy compared to RCC pre-surgery
Hauser S. (2012, Germany) [15]	240 (117 RCC, 14 benign renal tumor; 109 healthy)	miR-26a-2*, miR-191, miR-337-3p and miR-378	miR-378 equally expressed in RCC, benign renal tumor and controls
Heinemann F. G. (2018, Germany) [16]	169 (86 ccRCC, 55 benign renal tumor; 28 healthy)	miR-122-5p, miR-206, miR-193a-5p	miR-122-5p (*p* = 0.002), miR-206 (*p* < 0.001) ↓ in ccRCC compared to controls; miR-193a-5p not statistically different in RCC, benign renal tumor and controls (all *p* > 0.3)
Huang G. (June 2020, China) [17]	256 (126 RCC; 130 healthy)	30 miRNAs evaluated; 8 miRNAs selected for validation (miR-149-5p, miR-224-5p, miR-34b-3p, miR-129-2-5p, miR-142-3p, miR-182-5p, miR-671-5p, miR-625-3p)	miR-224-5p (*p* < 0.05) and miR-149-5p (*p* < 0.05) ↑ in RCC; miR-34b-3p (*p* < 0.05), miR-129-2-3p (*p* < 0.05) and miR-182-5p (*p* < 0.05) ↓ in RCC; no statistical difference reported for miR-142-3p, miR-625-3p and miR-671-5p
Huang G. (July 2020, China) [18]	220 (110 RCC, 110 healthy)	miR-20b-5p, miR-30a-5p and miR-196a-5p	miR-20b-5p (*p* < 0.001), miR-30a-5p (*p* < 0.001) ↓ in RCC compared to controls; miR-196a-5p (*p* < 0.001) ↑ in RCC compared to controls
Iwamoto H. (2014, Japan) [19]	57 (34 ccRCC; 23 healthy)	miR-210	miR-210↑ in ccRCC compared to controls (*p* = 0.001)
Kalogirou C. (2020, Germany) [20]	100 (34 pRCC type 1, 33 pRCC type 2; 33 healthy)	let-7b, miR-10a-3p, miR-10b-5p, miR-21-5p, miR-126-3p, miR-127-3p, miR-142-3p, miR-155-5p, miR-199a-3p, miR-210-3p and miR-425-5p	No different expression of any serum miRNAs in pRCC compared to controls
Li M. (2017, China) [21]	278 (139 RCC; 139 healthy)	miR-22	miR-22 (*p* < 0.001) ↓ in RCC compared to controls; miR-22 (*p* < 0.001) ↑ in RCC post-nephrectomy compared RCC pre-surgery
Li R. (2022, China) [22]	220 (108 RCC; 112 healthy)	12 miRNAs evaluated; 6 miRNAs (miR-18a-5p, miR-138-5p, miR-141-3p, miR-181b-5p, miR-200a-3p and miR-363-3p) went through validation phase	Panel of 4 combined miRNAs (miR-18a-5p, miR-181b-5p, miR-138-5p and miR-141-3p) selected for RCC detection (AUC = 0.908)
Li R. (2023, China) [23]	224 (112 RCC; 112 healthy)	12 miRNAs evaluated; 6 miRNAs (miR-1-3p, miR-124-3p, miR-129-5p, miR-155-5p, miR-200b-3p and miR-224-5p) went through validation phase	miR-155-5p (*p* = 0.001) miR-224-5p (*p* < 0.001) ↑ in RCC compared to controls; miR-1-3p (*p* = 0.001), miR-124-3p (*p* = 0.003), miR-129-5p (*p* < 0.001) and miR-200b-3p (*p* < 0.001) ↓ in RCC compared to controls
Liu T.Y. (2016, China) [24]	64 (32 RCC; 32 healthy)	miR-210	miR-210 (*p* < 0.001) ↑ in RCC compared to controls
Liu Z. (2021, China) [25]	226 (113 ccRCC; 113 healthy)	miR-410	miR-410 (*p* < 0.001) ↑ in ccRCC compared to controls
Lou N. (2016, China) [26]	276 (106 ccRCC, 28 angiomyolipomas, 19 nccRCC; 123 healthy)	1523 miRNAs evaluated	miR-144-3p ↑ in ccRCC compared to angiomyolipomas and controls (both *p* < 0.0001); miR-144-3p ↓ in ccRCC post-nephrectomy compared to ccRCC pre-surgery (*p* = 7.02 × 10^−5^)
Redova M. (2012, Czech Republic) [27]	152 (105 ccRCC; 47 healthy)	667 miRNAs evaluated; 3 miRNAs (miR-378, miR-150, miR-451) selected for validation	miR-378 (*p* = 0.0003) ↑ in ccRCC compared to controls; miR-150 (*p* = 0.222); miR-451 (*p* < 0.0001) ↓ in ccRCC compared to controls
Teixeira A. L. (2014, Portugal) [28]	77 (43 RCC; 34 healthy)	miR-221, miR-222	miR-221 (*p* = 0.028) and miR-222 (0.044) ↑ in RCC compared to controls
Teixeira A. L. (2015, Portugal) [29]	577 (133 RCC; 443 healthy)	miR-7, miR-221, miR-222	miR-7 (*p* < 0.001), miR-221 (*p* = 0.035), miR-222 (*p* = 0.042) ↑ in RCC compared to controls
Tusong H. (2016, China) [30]	60 (30 RCC; 30 healthy)	miR-21 and miR-106a	miR-21 (*p* < 0.0001) and miR-106a (*p* < 0.0001) ↑ in RCC compared to controls; miR-21 (*p* < 0.0001) and miR-106a (*p* < 0.0001) ↓ in RCC 1 month after nephrectomy compared to RCC pre-surgery
Wang C. (2015, China) [31]	264 (132 ccRCC; 132 healthy)	754 miRNAs evaluated; 20 miRNAs went through validation phase	miR-193a-3p (*p* < 0.0001), miR-362 (*p* < 0.0001), miR-572 (*p* < 0.0001), miR-425-5p (*p* = 0.0480) and miR- 543 (*p* = 0.0405) ↑ in ccRCC compared to controls; miR-28-5p (*p* = 0.0010) and miR-378 (*p* = 0.0033) ↓ in ccRCC compared to controls; miR-382, miR-208b, miR-337-5p, miR-1300, miR-7, miR-194, miR-324-5p, miR-886-3p, miR-1225-3p, miR-663b, miR-1247, miR-520c-3p, miR-1208 no statistical difference between ccRCC and controls (*p* > 0.05)
Wang X. (2016, China) [32]	67 (57 ccRCC; 10 healthy)	miR-182	miR-182 (*p* < 0.05) ↓ in ccRCC compared to controls
Wen Z. (2024, China) [33]	224 (112 RCC; 112 healthy)	12 miRNAs evaluated; 8 miRNAs (miR-1-3p, miR-129-5p, miR-141-3p, miR-146b-5p, miR-187-3p, miR-200b-5p, miR-200a-3p and miR-486-5p) went through validation phase	miR-1-3p (*p* < 0.001), miR-129-5p (*p* < 0.001), miR-187-3p (*p* < 0.001) and miR-200a-3p (*p* < 0.001) ↓ in RCC compared to controls; miR-146b-5p (*p* < 0.01) ↑ in RCC compared to controls; miR-141-3p, miR-200b-5p and miR-486-5p no statistical difference between RCC and controls (*p* > 0.05)
Wulfken L. M. (2011, Germany) [34]	265 (108 ccRCC, 10 pRCC, 3 chRCC, 2 sRCC; 129 healthy; 3 angiomyolipoma; 10 oncocytoma)	318 miRNAs evaluated; 7 miRNAs (miR-106b*, miR-1233, miR-1290, miR-210, miR-7-1*, miR-320b and miR-93) went through verification; miR-1233 went through validation	miR-1233 (*p* = 0.044) ↑ in RCC compared to controls
Yadav S. (2017, India) [35]	45 (30 RCC; 15 healthy)	miR-34a, miR-141, miR-200c, miR-1233, miR-21-2	miR-34a (*p* < 0.001) and miR-141 (*p* = 0.003) ↓ in RCC compared to controls; miR-1233 (*p* < 0.001) ↑ in RCC compared to controls; miR-200c (*p* = 0.086) and miR-21-2 (*p* = 0.331) not differentially expressed in RCC and controls
Zhang Q. (2015, China) [36]	101 (82 ccRCC; 19 healthy)	miR-183	miR-183 (*p* < 0.01) ↑ in ccRCC compared to controls
Zhao A. (2013, France) [37]	110 (68 RCC; 42 controls)	miR-210	miR-210 (*p* < 0.0001) ↑ in ccRCC compared to controls; miR-210 ↓ in ccRCC after nephrectomy compared to ccRCC pre-surgery (*p* = 0.001)

**Table 2 cancers-17-00816-t002:** Risk of bias and certainty assessment of included studies.

	Risk of Bias (ROBINS-E)								Certainty Assessment (GRADE)			
Study (First Author, Publication Year)	Domain 1	Domain 2	Domain 3	Domain 4	Domain 5	Domain 6	Domain 7	Overall Risk of Bias	Inconsistency	Indirectness	Imprecision	Quality
Chanudet E. (2017) [12]	Low-risk	Low-risk	Low-risk	Low-risk	Low-risk	Low-risk	Low-risk	Low-risk	Not serious	Not serious	Serious ^1^	Moderate
Chen X. (2021) [13]	Low-risk	Low-risk	Low-risk	Low-risk	Low-risk	Low-risk	Low-risk	Low-risk	Not serious	Not serious	Not serious	High
Fedorko M. (2015) [14]	Low-risk	Low-risk	Low-risk	Low-risk	Low-risk	Low-risk	Low-risk	Low-risk	Not serious	Not serious	Not serious	High
Hauser S. (2012) [15]	Some concerns ^2^	Low-risk	Low-risk	Low-risk	Some concerns ^3^	Low-risk	Low-risk	Some concerns	Not serious	Not serious	Not serious	Moderate
Heinemann F. G. (2018) [16]	Some concerns ^2^	Low-risk	Low-risk	Low-risk	Low-risk	Low-risk	Some concerns ^4^	Some concerns	Not serious	Not serious	Serious ^1^	Low
Huang G. (June 2020) [17]	Some concerns ^2^	Low-risk	Low-risk	Low-risk	Low-risk	Low-risk	Low-risk	Some concerns	Not serious	Not serious	Not serious	Moderate
Huang G. (July 2020) [18]	Some concerns ^2^	Low-risk	Low-risk	Low-risk	Low-risk	Low-risk	Low-risk	Some concerns	Not serious	Not serious	Not serious	Moderate
Iwamoto H. (2014) [19]	Some concerns ^2^	Low-risk	Low-risk	Low-risk	Low-risk	Low-risk	Low-risk	Some concerns	Not serious	Not serious	Serious ^1^	Low
Kalogirou C. (2020) [20]	Some concerns ^2^	Low-risk	Low-risk	Low-risk	Low-risk	Low-risk	Low-risk	Some concerns	Not serious	Not serious	Serious ^1^	Low
Li M. (2017) [21]	Some concerns ^2^	High-risk ^5^	Low-risk	Low-risk	Low-risk	Low-risk	Low-risk	High-risk	Not serious	Serious ^6^	Not serious	Very low
Li R. (2022) [22]	Some concerns ^2^	Low-risk	Low-risk	Low-risk	Low-risk	Low-risk	Low-risk	Some concerns	Not serious	Not serious	Not serious	Moderate
Li R. (2023) [23]	Some concerns ^2^	Low-risk	Low-risk	Low-risk	Low-risk	Low-risk	Low-risk	Some concerns	Not serious	Not serious	Not serious	Moderate
Liu T.Y. (2016) [24]	Some concerns ^2^	Low-risk	Low-risk	Low-risk	Low-risk	Low-risk	Low-risk	Some concerns	Not serious	Not serious	Serious ^1^	Low
Liu Z. (2021) [25]	Some concerns ^2^	Low-risk	Low-risk	Low-risk	Low-risk	Low-risk	Low-risk	Some concerns	Not serious	Not serious	Not serious	Moderate
Lou N. (2017) [26]	Some concerns ^2^	Low-risk	Low-risk	Low-risk	Low-risk	Low-risk	High-risk ^7^	High-risk	Not serious	Not serious	Not serious	Low
Redova M. (2012) [27]	Some concerns ^2^	Low-risk	Low-risk	Low-risk	Low-risk	Low-risk	Low-risk	Some concerns	Not serious	Not serious	Serious ^1^	Low
Teixeira A. L. (2014) [28]	Low-risk	Low-risk	Low-risk	Low-risk	Low-risk	Low-risk	Low-risk	Low-risk	Not serious	Not serious	Serious ^1^	Moderate
Teixeira A. L. (2015) [29]	Low-risk	Low-risk	Low-risk	Low-risk	Low-risk	Low-risk	Low-risk	Low-risk	Not serious	Not serious	Not serious	High
Tusong H. (2017) [30]	Low-risk	Low-risk	Low-risk	Low-risk	Low-risk	Low-risk	Low-risk	Low-risk	Not serious	Not serious	Serious ^1^	Moderate
Wang C. (2015) [31]	Some concerns ^2^	Low-risk	Low-risk	Low-risk	Low-risk	Low-risk	Low-risk	Some concerns	Not serious	Not serious	Not serious	Moderate
Wang X. (2016) [32]	Some concerns ^2^	Low-risk	Low-risk	Low-risk	Low-risk	Low-risk	Low-risk	Some concerns	Not serious	Not serious	Very serious ^8^	Very low
Wen Z. (2024) [33]	Some concerns ^2^	Low-risk	Low-risk	Low-risk	Low-risk	Low-risk	Low-risk	Some concerns	Not serious	Not serious	Not serious	Moderate
Wulfken L. M. (2011) [34]	Some concerns ^2^	Low-risk	Low-risk	Low-risk	Low-risk	Low-risk	Low-risk	Some concerns	Not serious	Not serious	Not serious	Moderate
Yadav S. (2017) [35]	Some concerns ^2^	Low-risk	Low-risk	Low-risk	Low-risk	Low-risk	Low-risk	Some concerns	Not serious	Not serious	Serious ^1^	Low
Zhang Q. (2015) [36]	Some concerns ^2^	Low-risk	Low-risk	Low-risk	Low-risk	Low-risk	Some concerns ^9^	Some concerns	Not serious	Not serious	Serious ^1^	Low
Zhao A. (2013) [37]	Low-risk	Low-risk	Low-risk	Low risk	Low-risk	Low-risk	Low-risk	Low-risk	Not serious	Not serious	Serious ^1^	Moderate

^1^ Small cohort (<200 participants between patients and controls considering all phases of the study). ^2^ No confounding factors mentioned. ^3^ Missing data. ^4^ miRNA selected without explanation. ^5^ No methods reported regarding serum miRNA. ^6^ Study mainly on miRNA expression in tissue samples. ^7^ No explanation for selected results. ^8^ Small cohort and no confidence intervals reported. ^9^ No sensitivity/specificity report.

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
