# Peer review of "An Evaluation of Serum miRNA in Renal Cell Carcinoma: A Systematic Review"

_cancers, 2025, doi:10.3390/cancers17050816_

Round 1

Reviewer 1 Report

Comments and Suggestions for Authors

The current review is the great topic on the miRNA distribution in blood plasma. the authors collect data from various sources and classify them using different parameters. this is very timely but needs some more modifications. 

pls add one more figures on the most recurring sequences of miRNA and if they have any disease relevance. this part could be added to the manuscript to make it complete. 

also some miRNA could be very low in detection and could only be detected at certain time points. this part also needs to be elaborated. 

Author Response

Comment 1: pls add one more figures on the most recurring sequences of miRNA and if they have any disease relevance. this part could be added to the manuscript to make it complete. 

Response 1: thank you for your suggestions, we added a graphic (Figure 2) to help readers visualize the expression of all studied miRNAs.

Comment 2: also some miRNA could be very low in detection and could only be detected at certain time points. this part also needs to be elaborated. 

Response 2: We added this sentence to the Discussion: “Many studies did not find any relevant expression of miRNAs in patients’ serum. This could be due to miRNAs fluctuations or laboratory equipment not able to detect lower blood concentrations of miRNAs.”

Reviewer 2 Report

Comments and Suggestions for Authors

This systematic review focuses on 26 studies  providing  a detailed examination of

circulating miRNAs as potential diagnostic biomarkers for RCC. 

The approach  applied by the Authors to systematic review seems  clear and adequate in method; though, some aspects are worthy of deeply explanation. 

1) Authors  decided not to perform metanalysis because they  expected excessive heterogeneity between studies in matter of type of miRNA, extraction method and normalizers used. This quote needs to be supported by data.

2) A  detailed examination of circulating miRNAs as potential diagnostic biomarkers for RCC. The included studies exhibited marked heterogeneity in terms of their design, target miRNAs, normalizers used during stabilization, and RCC subtypes studied (ccRCC, pRCC, or unspecified). This variability significantly limited the comparability of findings and the potential for quantitative synthesis. Also aspect inherent laboratory investigations have to be considered in systematic review regarding miRNA.

3) Among the investigated miRNAs, miR-210 emerged as a central focus, with most studies demonstrating its overexpression in RCC patients compared to controls. Conversely, conflicting results for miRNAs such as miR-21-5p and miR-378, which showed limited or inconsistent differentiation between RCC and controls, highlighted the variability in miRNA behavior across different studies and settings. This inconsistency may reflect differences in study populations, disease stages, or technical approaches, emphasizing the need for standardized protocols. Several studies proposed miRNA panels to enhance diagnostic accuracy [12,21,30,32]. The authors identified multi-miRNA combinations that could outperform individual biomarkers in sensitivity and specificity. Such panels may address the limitations of single-marker variability and improve robustness in diverse patient populations. Some papers evaluated dynamic changes post-surgery suggesting their potential utility as markers for both diagnosis and treatment monitoring. Our findings suggest that specific miRNAs could aid not only in diagnosis but also in assessing disease progression and guiding therapeutic decisions. 

Data by Authors not explain enough limitation of the article, on the contrary  it is required by systematic review.

4) This systematic review represents the most comprehensive and up-to-date analysis

of circulating miRNAs as diagnostic biomarkers for RCC to date. It builds upon and expands the scope of earlier reviews by incorporating a larger number of studies, reflecting the growing body of research in this field. In contrast to previous reviews, which often included a broader spectrum of RNA types or focused on tissue-derived miRNAs, our analysis exclusively examines serum-derived circulating miRNAs [37,38].

Authors should explain in details differences with serum-derived miRNAs, for knowledge of readers.

5) Consistent with previous reviews, our analysis identifies miR-210 as a consistently overexpressed biomarker in RCC, reinforcing findings from earlier studies [37]. However, discrepancies among individual studies underscore the need for standardized protocols and larger validation cohorts to address variability in study designs and populations.

While previous studies primarily focused on individual miRNAs, our review included also studies evaluating the diagnostic role of miRNAs panel emphasizing the value of combining multiple markers to address the inherent variability and improve sensitivity and specificity.  Some more indication on   miR-210: as the biomarker in  molecular biology mir-210 microRNA is a short RNA molecule. MicroRNAs function to regulate the expression levels of other genes by several mechanisms; mir-210 has been strongly linked with the hypoxia pathway, and is upregulated in response to Hypoxia-inducible factors. It is also overexpressed in cells affected by cardiac disease and tumours (Adrenocortical carcinoma,Breast cancer, pancreatic, neck and head cancer), in which is it   known higher expression indicates lower probability for survival in patients with breast cancer Lánczky A, Nagy Á, Bottai G, Munkácsy G, Szabó A, Santarpia L, GyÅ‘rffy B (December 2016). "miRpower: a web-tool to validate survival-associated miRNAs utilizing expression data from 2178 breast cancer patients". Breast Cancer Research and Treatment. 160 (3): 439–446.) 

MiRNA-210 in particular, has been studied for its effects in rescuing cardiac function after myocardial infarcts via the up-regulation of angiogenesis and inhibition of cardiomyocyte apoptosis. Vella, R.; Pizzocaro, E.; Bannone, E.; Gualtieri, P.; Frank, G.; Giardino, A.; Frigerio, I.; Pastorelli, D.; Gruttadauria, S.; Mazzali, G.; et al. Nutritional Intervention for the Elderly during Chemotherapy: A Systematic Review. Cancers 2024, 16, 2809. https://doi.org/10.3390/cancers16162809.

6) The conclusions of this paper highlights importance of the targeted biomarker: 

Technological advances: innovations in miRNA detection, including next-generation sequencing and machine learning-based analysis, could improve sensitivity and specificity, making miRNA-based diagnostics more viable in routine clinical settings. The main limitations of this review included the limited number of databases considered for study selection and to evaluate only the miRNAs involved in RCC studies, excluding the potential association with other histological types. Moreover, a metanalysis could provide a better understanding of potential biomarkers for RCC once a higher homogeneity among studies will be reached.

Almost in my opinion, advantages, potential utility and standards approach should be emphasized in details.

Author Response

Comment 1: Authors  decided not to perform metanalysis because they  expected excessive heterogeneity between studies in matter of type of miRNA, extraction method and normalizers used. This quote needs to be supported by data.
Response 1: We added a visual summary of the studied miRNAs (Figure 2) and laboratory findings (Supplementary table 4) of all included studies to further highlight and support their heterogeneity.

Comment 2: A  detailed examination of circulating miRNAs as potential diagnostic biomarkers for RCC. The included studies exhibited marked heterogeneity in terms of their design, target miRNAs, normalizers used during stabilization, and RCC subtypes studied (ccRCC, pRCC, or unspecified). This variability significantly limited the comparability of findings and the potential for quantitative synthesis. Also aspect inherent laboratory investigations have to be considered in systematic review regarding miRNA.
Response 2: Thank you for your suggestion, we added this sentence to the Discussion after the reported quote: “Discrepancies can also be found in laboratory investigations such as different methods of miRNAs’ extractions, normalizer used and quantitative analysis.”

Comment 3: Among the investigated miRNAs, miR-210 emerged as a central focus, with most studies demonstrating its overexpression in RCC patients compared to controls. Conversely, conflicting results for miRNAs such as miR-21-5p and miR-378, which showed limited or inconsistent differentiation between RCC and controls, highlighted the variability in miRNA behavior across different studies and settings. This inconsistency may reflect differences in study populations, disease stages, or technical approaches, emphasizing the need for standardized protocols. Several studies proposed miRNA panels to enhance diagnostic accuracy [12,21,30,32]. The authors identified multi-miRNA combinations that could outperform individual biomarkers in sensitivity and specificity. Such panels may address the limitations of single-marker variability and improve robustness in diverse patient populations. Some papers evaluated dynamic changes post-surgery suggesting their potential utility as markers for both diagnosis and treatment monitoring. Our findings suggest that specific miRNAs could aid not only in diagnosis but also in assessing disease progression and guiding therapeutic decisions. 

Data by Authors not explain enough limitation of the article, on the contrary  it is required by systematic review.

Response 3: We implemented the limitation of the review adding this sentence at the end of the Discussion section: “Although our review was in line with the current state of available evidence, considering the large number of existing miRNAs, few papers on different miRNAs are not enough to determine strong biomarkers for RCC. We suspected that this number could be underestimated because of publication bias.

Comment 4: This systematic review represents the most comprehensive and up-to-date analysis of circulating miRNAs as diagnostic biomarkers for RCC to date. It builds upon and expands the scope of earlier reviews by incorporating a larger number of studies, reflecting the growing body of research in this field. In contrast to previous reviews, which often included a broader spectrum of RNA types or focused on tissue-derived miRNAs, our analysis exclusively examines serum-derived circulating miRNAs [37,38].

Authors should explain in details differences with serum-derived miRNAs, for knowledge of readers.

Response 4: As you suggested, we added a brief paragraph with new reference that explains more clearly what could be the benefits of researching miRNAs in tissue or serum following the paragraph quoted above in the Discussion section: “Similarly to miRNAs expressed in tissue, dysregulated expression of serum miRNA might play a role in the generation or maintenance of solid tumors [40].  While investigating miRNAs expressed in tissue could help better understand their role in cancer generation, proliferation or metastatic process, serum miRNA could offer a viable noninvasive diagnostic marker for earlier cancer detection.”

Comment 5: Consistent with previous reviews, our analysis identifies miR-210 as a consistently overexpressed biomarker in RCC, reinforcing findings from earlier studies [37]. However, discrepancies among individual studies underscore the need for standardized protocols and larger validation cohorts to address variability in study designs and populations.

While previous studies primarily focused on individual miRNAs, our review included also studies evaluating the diagnostic role of miRNAs panel emphasizing the value of combining multiple markers to address the inherent variability and improve sensitivity and specificity.  Some more indication on   miR-210: as the biomarker in  molecular biology mir-210 microRNA is a short RNA molecule. MicroRNAs function to regulate the expression levels of other genes by several mechanisms; mir-210 has been strongly linked with the hypoxia pathway, and is upregulated in response to Hypoxia-inducible factors. It is also overexpressed in cells affected by cardiac disease and tumours (Adrenocortical carcinoma,Breast cancer, pancreatic, neck and head cancer), in which is it   known higher expression indicates lower probability for survival in patients with breast cancer Lánczky A, Nagy Á, Bottai G, Munkácsy G, Szabó A, Santarpia L, GyÅ‘rffy B (December 2016). "miRpower: a web-tool to validate survival-associated miRNAs utilizing expression data from 2178 breast cancer patients". Breast Cancer Research and Treatment. 160 (3): 439–446.) 

MiRNA-210 in particular, has been studied for its effects in rescuing cardiac function after myocardial infarcts via the up-regulation of angiogenesis and inhibition of cardiomyocyte apoptosis. Vella, R.; Pizzocaro, E.; Bannone, E.; Gualtieri, P.; Frank, G.; Giardino, A.; Frigerio, I.; Pastorelli, D.; Gruttadauria, S.; Mazzali, G.; et al. Nutritional Intervention for the Elderly during Chemotherapy: A Systematic Review. Cancers 2024, 16, 2809. https://doi.org/10.3390/cancers16162809.

Response 5: Thank you for your suggestions, we added an informative segment on miR-210 in the Discussion section following the studies referenced in your comment: “Through a variety of processes, microRNAs control the expression levels of other genes, such as elevated levels of miR-210 in response to hypoxia-inducible factors and its association with the hypoxia pathway. Additionally, miR-210 overexpression in tissues affected by heart illness and tumors (adrenocortical carcinoma, breast cancer, pancreatic, neck and head cancer) is known to be associated with a decreased chance of survival[41]. Moreover, miR-210 has been investigated for its potential to restore heart function following myocardial infarcts by inhibiting cardiomyocyte death and upregulating angiogenesis[42]; this could further suggest miR-210 role in tumoral neoangiogenesis.”

Comment 6: The conclusions of this paper highlights importance of the targeted biomarker: 

Technological advances: innovations in miRNA detection, including next-generation sequencing and machine learning-based analysis, could improve sensitivity and specificity, making miRNA-based diagnostics more viable in routine clinical settings. The main limitations of this review included the limited number of databases considered for study selection and to evaluate only the miRNAs involved in RCC studies, excluding the potential association with other histological types. Moreover, a metanalysis could provide a better understanding of potential biomarkers for RCC once a higher homogeneity among studies will be reached.

Almost in my opinion, advantages, potential utility and standards approach should be emphasized in details.

Response 6: We added two bullet point that explain why these suggestions could improve future research:

  • “An international consensus on laboratory investigations for miRNA extraction, profiling, stabilization and quantitative analysis could provide a clearer interpretation of results.
  • Institution of a global updated library of miRNA that could help researchers explore not yet investigated miRNAs and consolidate international findings.”

Reviewer 3 Report

Comments and Suggestions for Authors

Cochetti G and Guadagni L et al. reported an interesting systematic review about serum miRNA in RCC. The topic was important, and fell within the scope of Cancers. The reviewer suggested a Minor Revision for this paper. Detailed comments:

1-     Please delete RCC as a Keyword. Some of the audience might be unfamiliar with this abbreviation, and it added no value.

2-     The research questions (RQs) of this systematic review should be described at the end of the Introduction.

3-     There were papers included in the review. Please discuss whether this number was appropriate.

4-     For Table 2: Could the level of risk be more quantitatively analyzed, rather than qualitative text?

5-     Was it necessary to cite Supplementary Materials of published papers and link to software, as references?

Author Response

Comment 1: Please delete RCC as a Keyword. Some of the audience might be unfamiliar with this abbreviation, and it added no value.

Response 1: Thank you for your suggestion. We deleted the keyword.

Comment 2: The research questions (RQs) of this systematic review should be described at the end of the Introduction.

Response 2: To clarify our RQ, we added this sentence at the end of the introduction as you suggested: “Our primary research question (RQ) is: What serum miRNA are differentially expressed in adult patients with RCC compared to healthy individuals?
Our secondary RC is: What serum miRNA express different behavior before and after surgery in RCC patients?

Comment 3: There were papers included in the review. Please discuss whether this number was appropriate.

Response 3: We added a sentence at the end of the Discussion section to address the number of included papers as a limitation: “Although our review was in line with the current state of available evidence, considering the large number of existing miRNAs, few papers on different miRNAs are not enough to determine strong biomarkers for RCC. We suspected that this number could be underestimated because of publication bias.”

Comment 4: For Table 2: Could the level of risk be more quantitatively analyzed, rather than qualitative text?

Response 4: The ROBINS-E offers a qualitative analysis and is considered to be one of the most comprehensive tools for risk of bias assessment for diagnostic accuracy studies. The authors think that reporting the risk domain quantitatively (e.g. Study 1 – Domain 1: 5/6) does not allow to report equally all results since some secondary questions for each domain have to be addressed only if the first questions have certain answers. This could lead to different denominators for different studies in the same domain. The qualitative assessment is standardized and allows the reader to compare all included papers.

Comment 5: Was it necessary to cite Supplementary Materials of published papers and link to software, as references?

Response 5: Where the extracted data was found in the supplementary material of the papers, we added the link to the supplementary material in the reference of the said paper in the Reference section. We also added the reference for the software used in the literature research phase.